# Persimmon Proanthocyanidins with Different Degrees of Polymerization Possess Distinct Activities in Models of High Fat Diet Induced Obesity

**DOI:** 10.3390/nu14183718

**Published:** 2022-09-09

**Authors:** Ying Yu, Ping Chen, Xiaofang Li, Shanshan Shen, Kaikai Li

**Affiliations:** 1College of Food Science and Technology, Key Laboratory of Environment Correlative Food Science, Huazhong Agricultural University, Wuhan 430070, China; 2Wuhan Children’s Hospital, Tongji Medical College, Huazhong University of Science & Technology, Wuhan 430016, China

**Keywords:** proanthocyanidins, polymerization degree, digestion, anti-obesity ability

## Abstract

Proanthocyanidins is a kind of polyphenol that had been found with strong prevention ability on high fat diet induced obesity. However, whether proanthocyanidins with different polymerization degree showed different anti-obesity ability is unclear. Therefore, in this study, the effects of persimmon proanthocyanidins (P-PCs) and persimmon oligo-proanthocyanidins (P-OPCs) on high-fat diet induced obesity were systematically investigated. The findings indicated that both of P-PCs and P-OPCs significantly reduced the body weight, and P-PCs showed stronger anti-obesity ability compared with P-OPCs, P-OPCs seemed with stronger ability on improvement of insulin resistance. Furthermore, gut microbiota results indicated that the composition of the gut microbiota was changed after P-PCs and P-OPCs intervention in C57BL/6J mice. In addition, P-PCs exhibited strong inhibitory on the digestion of starch and fat. Above all, this study indicated that P-PCs showed stronger anti-obesity ability compared with P-OPCs.

## 1. Introduction

Obesity and its associated metabolic syndrome have become an important global public health problem [1,2,3,4]. Until now, developing effective innovative interventions is still an urgent issue. Application of some digestive enzyme inhibitors, which could prevent the absorption of nutrients, seems a useful strategy to alleviate obesity. Several naturally phytochemicals found in foods have been found with strong digestion enzyme inhibitory activities; for example polyphenols, which showed potential anti-obesity ability [5,6]. Proanthocyanidins (PCs) are kinds of polyphenols existing in many plant foods, and flavan-3-ols is the main structural unit which with various degrees of polymerization (DP) [7,8]. PCs have been found with many biological activities such as antioxidative, anti-obesity, antibacterial potential [6,8,9]. It has been found that PCs could inhibit the digestion enzyme, reshape intestinal microenvironment and prevent adipocyte differentiation, which played key roles for its anti-obesity potential.

Depending on the DP, PCs are classified as polymeric proanthocyanidins (P-PCs, DP > 5) and oligomeric proanthocyanidins (OPCs, DP 2 to 5). Previous results have found that PCs with DP > 4 could not be absorbed [10]. More importantly, it has been proven that the biological activities and bioavailability of PCs was directly influenced by the degrees of polymerization. For example, Pierini et al. found that PCs with different DP exhibited different anti-cancer ability in various cell lines [11]. Quiñones et al. found that OPCs with anti-hypertensive effect through adjusting the endothelium-derived NO bioavailability [12]. However, there were also other results, which indicated P-PCs with strong activities. Bitzer et al. found that P-PCs from cocoa showed the strongest effective on gut barrier function and epithelial inflammation [13]. Gonçalves et al. also found that with the increase of the degree of polymerization, the inhibitory potential of α-amylase was increased; a high degree of polymerization made it more easy to form a stable interaction with a-amylase [14]. These findings highlighted the fact that the relationship between degrees of polymerization of PCs and their bio-activities were highly context dependent. However, until now, how the degrees of polymerization influence the anti-obesity of PCs is still unclear.

Persimmon is a widely consumed fruit with a high content of PCs [15]. A large number of studies have demonstrated persimmon PCs with various bio-activities, for example enzyme inhibitory ability, antioxidant activity, anti-obesity, and anti-inflammatory activity [15,16]. However, persimmon PCs were a kind of PC with high degree of polymerization (mDP = 26), which was insoluble in water and with a strong astringent taste; in the meanwhile, it was easy to connect with protein and polysaccharide, which seriously limited the usages of P-PCs [17]. In our previous study, we prepared P-OPCs from P-PCs with catalytic hydrogenation; the average DP of P-OPCs was 2.8 [16]. P-OPCs showed stronger solubility, antibacterial activities and free radical scavenging potential compared with P-PCs [16,18]. Thus, it is of interest to determine whether P-PCs and P-OPCs demonstrate different anti-obesity activities. Therefore, in this study, the potential effects of P-PCs and P-OPCs on high-fat diet induced obesity were investigated.

## 2. Materials and Methods

### 2.1. Catalytic Hydrogenolysis of P-PCs

Ripe persimmon fruit “GongchengYueshi” (*Diospyros kaki* L.) were obtained from a local market (Baishazhou Market, Wuhan city, China). P-PCs were extracted from whole fruit as previous described [15]. The preparation of P-OPCs was performed with catalytic hydrogenolysis method [16]. The hydrogenolysis condition was at 3 MPa, 120 °C, 3 h with 0.4 mg/mL of Pd/C; the average DP of P-PCs and obtained P-OPCs were 26 and 2.8 [16]. The detail information about the composition of oligomers of P-OPCs was as our previous study described [16].

### 2.2. In Vivo Animal Study

#### 2.2.1. Animals and Dosage Regimen

All experiments were confirmed by the Laboratory Animal Research Center and the Ethics Committee of Huazhong Agricultural University. Male C57BL/6J mice (18–20 g) were kept in a stable environment (20 ± 2 °C, 12-h light/dark cycles) with free access to water and food. D12450B and D12492 were purchased from Readydietech. Co., Ltd. (Shenzhen, China). The detail information of the formula of D12450J and D12492 could be found in Appendix A. C57BL/6J mice were divided into four groups (*n* = 10): (1) Control group (C); (2) the High Fat Diet (HFD) group (M); (3) the HFD gavaged with 40 mg kg^−1^ of P-PCs group (PPCs); and (4) the HFD gavaged with 40 mg kg^−1^ of P-OPCs group (OPCs). Food intake and body weights were recorded twice a week.

#### 2.2.2. Glucose Tolerance Tests (GTTs)

The Oral GTTs were investigated during week 9 in all four groups. Mice were fasted overnight and then gavaged with 2 g glucose/kg BW of glucose. At glucose administration, tail blood samples were collected at 0, 30, 60, 90, and 120 min, and the blood glucose content was measured with a blood glucose meter (Roche Diagnostics, Germany), as in our previous study.

#### 2.2.3. Tissue Sample Collection

Animals were fasted overnight and anesthetized after the treatment. An eyeball blood sample was first collected, and then centrifuged at 3000 rpm for 10 min to prepare serum samples, then frozen at −80 °C. Livers and fat pad tissue were collected and weighed, then the tissue samples were cut into small pieces for fixed in 4% paraformaldehyde or stored at −80 °C for further analysis.

#### 2.2.4. Biochemical Analyses and Histology

Serum triglycerides (TGs), total cholesterol (TC), low density lipoprotein (LDL), and high density lipoprotein (HDL) levels were investigated using enzymatic colorimetric methods with special reagent kits (Nanjing Jiancheng, China) according to the manufacturers’ instructions. Fixed liver and adipose tissue were embedded in paraffin, then cut into sections with a thickness of 4 μm, and then stained with hematoxylin and eosin (H&E). Morphological changes were observed under a microscope.

### 2.3. DNA Extraction and Information Analysis of Gut Microbiota

After the mice were sacrificed, feces were quickly collected from the cecum and frozen in liquid nitrogen, then stored at −80 °C. Total microbial DNA was extracted using an E. Z.N.A™ Mag-Bind Soli DNA Kit (Omega Bio-tek, Norcross, GA, USA). The extracted DNA sample was accurately quantified using the Qubit3.0 DNA detection kit. The V3-V4 hypervariable region of the bacterial 16S rRNA gene were amplified with primer pairs (338F and 806R). Purified amplicons were paired-end sequence with an Illumina NovaSeq PE250 platform (Illumina, San Diego, CA, USA) by Majorbio Bio-Pharm Technology Co., Ltd. (Shanghai, China).

### 2.4. Enzyme Activity Assay

Assays of the inhibition of P-PCs and P-OPCs on activities of α-amylase and pancreatic lipase α-Amylase (Sigma-Aldrich, St. Louis, MO, USA) activity was investigated with dinitrosalicylic acid color reagent, pancreatic lipase (Sigma-Aldrich, St. Louis, MO, USA) activity was estimated using 4-MUO method as our previous report [5]. The 50% inhibition concentrations (IC_50_) of P-PCs and P-OPCs were evaluated.

#### Fluorescence Quenching Measurements

The influences of PCs and P-OPCs on fluorescence spectra of α-Amylase and pancreatic lipase were investigated with Hitachi F-4600 fluorescence spectrometer as our previous described [19]. α-Amylase and pancreatic lipase were both prepared in PBS at 1 mg/mL. α-Amylase or pancreatic lipase with or without P-PCs and P-OPCs (0, 10, 20, 30, and 40 mg/L) was hatched at 37 °C for 15 min. Fluorescence emission were recorded in a 1 cm quartz-cell at λex of 278 nm, and λem of 290 to 500 nm for α-Amylase, and λex of 280 nm, and λem of 300 to 500 nm for Pancreatic lipase. Both excitation and emission slit widths were set at 5 nm and maintained with a scanning rate of 1200 nm/min. The Stern–Volmer equation was applied to investigate the fluorescence quenching mechanism.

### 2.5. Statistical Analysis

The results were presented as Mean ± SDs. All data were analyzed with one-way ANOVA using GraphPad Prism ver. 5.1 software (La Jolla, CA, USA). Statistical difference was set at *p* < 0.05.

## 3. Results

### 3.1. P-PCs Administration Showed Stronger Potential on HFD Induced Weight Gain Compared with P-OPCs

As the results showed in Figure 1a, the food intake of the mice in C groups was higher than that of the M group; the PPCs group was significantly lower than the M group (Figure 1a). The mice in HFD group showed a higher body weight than that in C group (42.17 ± 4.65 vs. 29.38 ± 2.86 g). The groups that administered with both of P-PCs and P-OPCs exhibited significant inhibitory potential in body weight (33.05 ± 3.02 vs. 42.17 ± 4.65 g and 36.65 ± 4.03 vs. 42.17 ± 4.65 g for PPCs and POPCs, respectively) (Figure 1b,c). In the meanwhile, HFD treatment could induce the increase of serum TC, TG and LDL. P-PCs and P-OPCs treatment significantly reduced the levels of serum TGs (0.90 ± 0.16 vs. 1.27 ± 0.21 mmol L^−1^, *p* < 0.01) and TC (2.58 ± 0.57 vs. 4.16 ± 0.35 mmol/L, *p* < 0.01; 3.32 ± 0.38 vs. 4.16 ± 0.35 mmol/L, *p* < 0.05 for PPCs and POPCs, respectively) (Figure 1d,e), and P-PCs treatment group showed lower TC and Cho content compared with that of P-OPCs (*p* < 0.05). For LDL-C and HDL, P-PCs treatment could significantly reverse the change that induced by M (Figure 1f,g). For P-OPCs, even there was no significant different compared with the M treatment, P-OPCs treatment also exhibited improvement potential.

### 3.2. P-PCs and P-OPCs Administration Improved Insulin Resistance

As shown in Figure 2a, the mice in M group showed higher fasting glucose levels (4.76 ± 0.53 vs. 8.17 ± 1.02 mmol/L, *p* < 0.01), and treatment with P-PCs and P-OPCs showed a significant decrease (5.99 ± 0.63 vs. 8.17 ± 1.02 mmol/L; 5.87 ± 0.52 vs. 8.17 ± 1.02 mmol/L, for P-PCs and P-OPCs, respectively). Compared with that in C group, the blood glucose levels were much higher in the M group after glucose administration (Figure 2b,c). The area under the curve (AUC) of glucose also indicated that both of P-PCs and P-OPCs could improve insulin resistance (AUC: 34.86 ± 1.24 vs. 43.48 ± 2.23 and 32.70 ± 1.82 vs. 43.48 ± 2.23 for P-PCs and P-OPCs, respectively).

### 3.3. Dietary P-PCs and P-OPCs Reduce Lipid Accumulation in HFD Induced Obese Mice

As the results showed in Figure 3a, both P-PCs and P-OPCs could significantly reduce the size and number of lipid droplets observed in the livers and epididymal fat mass. HFD could induce the increase of the weight of inguinal fat mass, epididymal fat mass, and perirenal fat mass, and both P-PCs and P-OPCs could inhibit the expansion of adipose tissue, compared with P-OPCs; P-PCs showed stronger inhibitory potential (Figure 3b–d). In addition, as shown in Figure 3b, the epididymal fat mass weights of mice in PPCs group were significantly lower than that in P-OPCs group (Figure 3b). Both P-PCs and P-OPCs treatment could decrease the cell area of epididymal fat mass induced by HFD, and compared with P-OPCs, P-PCs should stronger inhibitory potential (Figure 3e). In the meanwhile, as shown in Figure 3f,g, P-PCs and P-OPCs significantly decreased the liver weight and inhibited the accumulation of liver lipids compared with M group.

### 3.4. Effect of P-PCs and P-OPCs on Gut Microbiota Diversity

Next, we further explore the impact P-PCs on the gut microbiota composition via 16S rRNA gene sequencing. There were 447, 402, 402, and 427 OTUs in the C group, M group, PPCs, and POPCs groups, respectively (Figure 4a). Accounting for 75% of the total, OTUs (313) were common existing among the four groups. For alpha diversity analysis, P-PCs treatment groups showed higher Shannon index and lower Simpson index compared with that in HFD group, which indicated that P-PCs treatment improved the alpha diversity decreased induced by M (Figure 4b–d). PCoA analysis of the four groups indicated that gut microbiota compositions were independent. The difference in the first (15.53%) and second (38.99%) principal component of gut microbiota indicated the difference in gut microbiota composition. Bray–Curtis assay showed the gut microbiota from C group, gut microbiota of M mice clustered separately with significant difference (Adonis, *p* = 0.001) (Figure 4e,f). The results showed that HFD intervention significantly changed the composition of the gut microbiota, however, there was no significant difference between P-PCs treatment groups and M group.

### 3.5. Composition of the Gut Microbiota

Phylum distributions of different treatment group were shown in Figure 5a, *Bacteroidetes* and *Firmicutes* were the primary bacteria (about +80%). The relative abundance of *Bacteroidetes* was significantly reduced in the M group (5.42 ± 0.39% vs. 17.46 ± 4.50%, *p* < 0.05), and *Firmicutes* was increased (79.38% ± 11.81% vs. 66.18 ± 5.88%, *p* < 0.05) compared with that in the C group. Compared with that in M group, P-PCs treatment could significantly increase the relative abundance of *Bacteroidetes* and *Epsilonbacteraeota* (*p* < 0.05). In P-OPCs treatment group, there was no significant difference compared with M; Bacteroidetes was also increased and *Proteobacteria* and *Actinobacteria* also decreased. There was no significant different between P-PCs and P-OPCs treatment groups (Figure 5b,c).

#### 3.5.1. Genus Level

As the results presented in Figure 6a–e show, the relative abundances of *Faecalibaculum*, *norank-f-Muribaculaceae*, and *unclassified f-Ruminococcaceae* showed significant reduction in the M group, whereas those of *Blautia* and *Lactobacillus* (*p* < 0.05) were significantly increased. In the PPCs and POPCs groups, the relative levels of *Helicobacter* and *Odoribacter* were significantly increased in the P-PCs group, while those of *Romboutsia* and *Ruminococcaceae* were significantly reduced compared with those in the HFD group. In the P-OPCs treatment group, the relative level of *Tyzzerella* was significantly increased, while those of *Romboutsia* was significantly reduced. Compared with the P-OPCs treatment group, the abundances of *Lactobacillus* was higher, and *unclassified_f_Family_XIII*, *Ruminococcaceae_UCG_009* and *Clostridium_sensu_stricto_1* were lower in the P-PCs treatment groups.

#### 3.5.2. Species Level

As the results presented in Figure 7a–e show, the relative abundances of *uncultured_bacterium_g_Faecalibaculum*, *unclassified_f_Ruminococcaceae*, and *uncultured_bacterium_g_norank_f_Muribaculaceae* were significantly reduced in the M group. The relative abundance of *unclassified_g_Blautia* and *unclassified_g_Lactobacillus* were significantly increased in the mice of M group. Compared with M group, the relative abundances of unclassified_g_Odoribacter, Helicobacter_hepaticus, and unclassified_g_Lachnospiraceae were significantly increased; and uncultured_bacterium_g_Romboutsia, uncultured_bacterium_g Ruminococcaceae_UGC-009 were reduced in P-PCs treatment group. The P-OPCs treatment significantly increased the levels of uncultured_bacterium_g_Roseburia and uncultured_bacterium_g_Tyzzerella, and the relative abundances of uncultured_bacterium_g_Romboutsia was significantly lower compared with that in the HFD group. There were also many differences between P-PCs and P-OPCs treatment groups; for example, unclassified_g__Lactobacillus. These findings suggested that HFD could destroy the structure of the gut microbiota, P-PCs and P-OPCs could improve the gut microbiota disorder induced by HFD.

### 3.6. Inhibitory Activities of P-PCs and P-OPCs on Digestive Enzymes Activities

The inhibitory potentials of P-PCs and P-OPCs on α-amylase and pancreatic lipase were investigated. Compared with P-OPCs, P-PCs showed stronger inhibitory potential on pancreatic lipase and α-amylase; with the IC50 values were 78.6 μg/mL (P-PCs) and 105.4 μg/mL (P-OPCs) for α-amylase, 22.9 μg/mL (P-PCs) and 43.1 μg/mL (P-OPCs) for pancreatic lipase. The significant inhibitory ability of P-PCs and P-OPCs on α-amylase and pancreatic lipase suggested that P-PCs and P-OPCs might directly bind to the enzymes. Therefore, the binding characters between P-PCs and P-OPCs and α-amylase or pancreatic lipase were further investigated.

The fluorescence quenching spectra of α-amylase by P-PCs and P-OPCs at 37 °C were shown in Figure 8. A sturdy fluorescence emanation peak at 345 nm was found in α-amylase after exciting at λ = 280 nm. For α-amylase, the fluorescence strength of α-amylase markedly diminished with the increase of the dose of P-PCs and P-OPCs; the EC_50_ values were 10.94 mg/L of P-PCs and 16.57 mg/L of P-OPCs at 37 °C. P-PCs and P-OPCs also dose-dependently quenched the fluorescence of pancreatic lipase, and the EC_50_ was 17.85 mg/L of P-PCs and 21.58 mg/L of P-OPCs at 37 °C. Compared with P-OPCs, P-PCs showed large Ksv and Kq values both with α-amylase and pancreatic lipase, which indicated a stronger interaction between P-PCs and the enzymes molecular. These results indicated that P-PCs showed stronger quenching potential on α-amylase and pancreatic lipase compared with P-OPCs. These results were also consisted with previous studies that the fluorescence quenching capacity of grape seed proanthocyanidins on α-amylase were positively collected to its DP [14].

## 4. Discussion

Proanthocyanidins exist in many food materials, for example, grape seed, persimmon, sorghum, and cocoa, mainly existing in the form of polymeric proanthocyanidins (>90%) [20,21]. In our previous study, the P-OPCs from P-PCs were prepared by the catalytic hydrogenation method, which demonstrates good solubility, bio-availability, and free radical scavenging ability. However, it has been found that the degrees of polymerization of proanthocyanidins played central role for the biological activities of PCs [6,14]. More importantly, the correlation between polymerization degree and bioactivity appears to be dependent on the different kinds of bioactivity. Therefore, in this study, the effects of P-PCs and P-OPCs on high-fat diet induced obesity were systematically investigated.

Nowadays, the rapidly increasing number of the population with obesity has become a serious health problem. In this study, a HFD induced obesity model was applied to systemically investigate the inhibitory properties of P-PCs and P-OPCs on diet induced obesity and the underlying mechanisms. The results indicated that both of P-PCs and P-OPCs reduced the body weight, TC, TG, and LDL, while increasing the content of HDL in HFD treatment mice. Compared with P-OPCs, P-PCs showed stronger anti-obesity potential, which showed lower body weight, TG, TC, and so on. However, in this study, there was an interesting phenomenon regarding the obesity-related insulin resistance; treatment with P-PCs and P-OPCs decreased the glucose levels, even there was no significant difference, and the P-OPCs treatment group showed lower fasting glucose level; meanwhile, GTT test also showed similar results. These results indicated that P-PCs and P-OPCs could improve insulin resistance, and P-OPCs seemed with stronger ability compared with P-PCs.

Many studies consistently supported dietary PCs with health benefits in murine models and humans; for example, anti-obesity, improve insulin sensitivity. It had been found that the improvement of PCs on insulin sensitivity were not linked to inhibitory potential in obesity [6]. Paquette et al. found that a mix rich in PACs and phenolic acids extract could increase insulin sensitivity and lower first-phase insulin secretion; however, this treatment showed no improvement effects on body weight, lipid profile, and oxidative stress [22]. Stull et al. also found the similar results [23]. Hokayem et al. found 2 g of grape polyphenols per day could prevent fructose induced insulin resistance [24]. These findings were consisted with our results, even P-OPCs with lower ability in inhibition of body weight, P-OPCs seems with stronger ability for insulin sensitivity compared with P-PCs treatment. These results also indicated that the mechanism of improvement of insulin sensitivity of P-PCs and P-OPCs were different.

Recently, growing evidence has indicated that diet induced obesity was correlated with the gut microbiota. High fat diet could induce gut microbiota disorder by reducing the diversity and destroying the microbiota structure. In this study, the Shannon index and the Simpson index indicated that the alpha diversity increased in the P-PCs and P-OPCs treatment groups, though not significantly different. PCoA analysis also found that composition of the gut microbiota was changed in HFD group; however, there was no significant change between PCs treatment groups and HFD group. It has been reported that HFD could alter cecum microflora populations and increase the abundance of *Firmicutes*, and decrease the abundance of *Bacteroidetes*. These results were similar in this study. Compared with HFD group, P-PCs and P-OPCs treatments could increase the abundance of Bacteroidetes. Compared with HFD group, both of P-PCs and P-OPCs treatments could decrease the abundance of Romboutsia. Wang et al. found that HFD could significantly induced the increase of Romboutsia, and walnut green husk extract treatment could significantly inhibit the obesity along with the decrease of Romboutsia, this result was consisted with our finding [25]. Though there were many significant differences between P-PCs and HFD, P-OPCs and HFD were found; however, there were only a few gut microbiota shown in the same manner in the P-PCs and P-OPCs treatments group compared with HFD group. Therefore, we hypothesized the anti-obesity ability of P-PCs and P-OPCs were not always dependent on the blooming or decreasing of any bacterium, despite being influenced by P-PCs and P-OPCs; the change in gut microbiota is not essential for P-PCs and P-OPCs to positively impact health markers.

Even though many studies found that gut microbiota was a key target of the biological ability of polyphenols, most of them, it seems, just described the results of the influences of polyphenols on gut microbiota; there was little research about why or how polyphenols regulate the structure of gut microbiota. It has been proved that the bioavailability of PCs was low when administered orally and accumulates in the intestine [26]; especially for polymeric PCs, it almost cannot be absorbed and, thus, reached the colon un-metabolized, where it could directly interact with the gut microbiota. However, PCs seem not be the optimum substrate for intestinal microorganism; in particular, PCs have been found with strong bacteriostatic activity. Therefore, we hypothesized that the anti-nutrition effects of the PCs played the central role for its anti-obesity ability. Firstly, PCs could inhibit the digestion and absorption of nutrients (starch, protein, and lipids), and then these undigested nutrients transited into the colon where they could be degradation and utilization as the substrate for intestinal microorganism, then modulated the gut microbiota. This hypothesis was also proven on acarbose, an α-Glucosidase inhibitor, which could be utilized to modulate the structure of gut microbiota to increase the butyrate content through shunting starch to the colon [27].

The absorption of proanthocyanidins was highly dependent on its degrees of polymerization and the linked type, even the OPCs could be directly absorbed in vivo animal models; however, the bioavailability was low [28]. Large amounts of PCs were catabolized by gut microflora to phenolic acids and other metabolites in the colon [29]. This suggested that these metabolites of PCs might be important compounds which could also play key roles to the improvements in glucose homeostasis. Wiese et al. found that the metabolism of PCs was also associated with its DP, the formation of 5-(3′,4′-dihydroxyphenyl)-valerolactone, a main metabolites of PCs, decreased with the increasing of DP [30]. In this study, persimmon tannin (P-PCs) was a kind of highly polymerized PC (mDP = 26), and P-OPC was a mixture of degradation products (mDP = 2.8) [16]. Compared with P-OPCs, P-PCs were hardly adsorbed. Therefore, especially for P-PCs, inhibiting the digestion of may play the central role for its anti-obesity ability. Thus, we further evaluated the inhibitory activities of P-OPCs and P-PCs on the main digestion enzymes. The results indicated that both of P-OPCs and P-PCs showed strong inhibitory potentials on these enzymes, and the P-PCs showed stronger inhibitory potential compared with P-OPCs. These results were also consisted with previous studies about the tannin from sorghum, which found that the inhibitory activity of condensed tannins on α-amylase was dependent on the DP [31]. Similar results were also found in procyanidins from cocoa extracts [32]. These findings could partly explain the stronger inhibitory ability of P-PCs on α-amylase and lipase compared with P-OPCs.

## 5. Conclusions

In this study, the results found P-PCs with showed strong anti-obesity ability compared with P-OPCs. Compared with P-PCs, P-OPCs seems with stronger ability on improvement of insulin resistance. Furthermore, the fecal microbiota results also indicated that both of PCs and P-OPCs changed the composition of the gut microbiota and modulated specific bacteria induced by HF diet. However, it also requires further work to investigate new strategies to decrease the limitation of applications of P-PCs in food processing.

## Figures and Tables

**Figure 1 nutrients-14-03718-f001:**
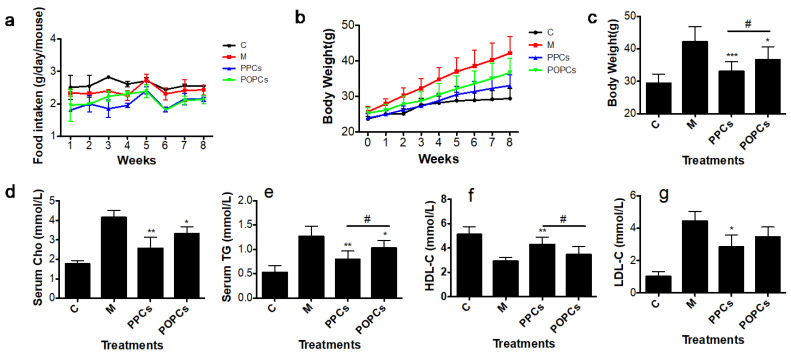
P-PCs and P-OPCs administration protect mice against excessive weight gain**.** C57BL/6 mice were fed a HFD with the administration of P-PCs and P-OPCs. (**a**) Food intake of mice with different treatment; (**b**) Body weight gain curve; (**c**) body weight at the last week; (**d**–**g**) Serum Cho, TG, HDL and LDL levels in different treatment groups. The results were analyzed with a one-way ANOVA with Tukey’s Multiple Comparison Test using GraphPad Prism ver. 5.1 software (San Diego, CA, USA). * *p* < 0.05, ** *p* < 0.01, *** *p* < 0.001; # *p* < 0.05 between PPCs and POPCs.

**Figure 2 nutrients-14-03718-f002:**
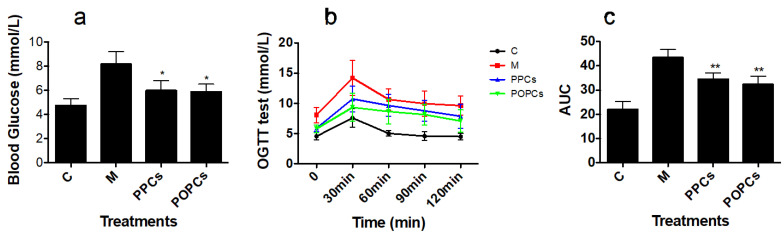
P-PCs and P-OPCs administration improve HFD induced insulin resistance. C57BL/6 mice were fed a HFD with the administration of P-PCs and P-OPCs. (**a**) Fasting blood glucose content; (**b**,**c**) GTT results after 8 weeks of P-PCs and P-OPCs administration. Blood glucose levels were measured at 0, 15, 30, 60, 90, and 120 min after oral administration of glucose (2 g kg^−1^). The results were analyzed with a one-way ANOVA with Tukey’s Multiple Comparison Test using GraphPad Prism ver. 5.1 software. * *p* < 0.05, ** *p* < 0.01.

**Figure 3 nutrients-14-03718-f003:**
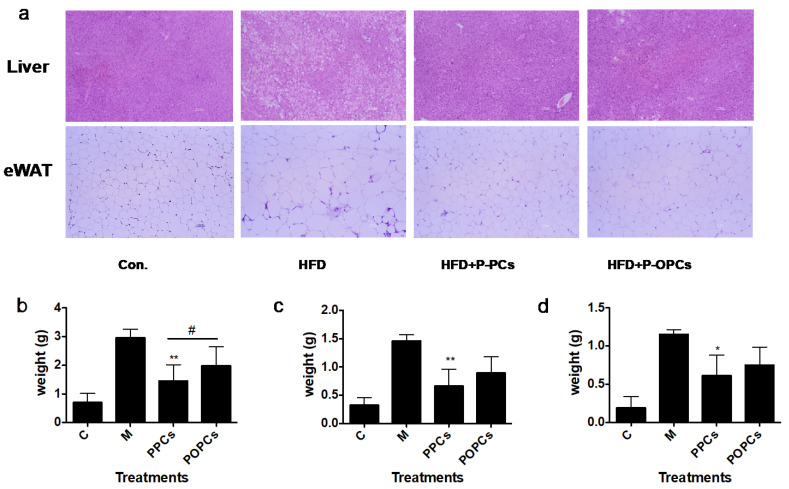
P-PCs and P-OPCs administration improves hepatic lipid accumulation and adipose tissue development. C57BL/6 mice were fed a HFD with the administration of P-PCs and P-OPCs. (**a**) H&E stained liver sections and epididymal adipose tissue; (**b**–**d**) weight of perirenal fat mass, epididymal fat mass and inguinal fat mass in different treatment groups; (**e**) Adipocyte size area of epididymal fat mass; (**f**,**g**) Liver weights and liver lipids accumulation in different group. Hepatic lipids were extracted by homogenizing the liver tissue in chloroform/methanol lipid extraction buffer and were analyzed according to the manufacturer’s protocol. The results were analyzed with a one-way ANOVA with Tukey’s Multiple Comparison Test using GraphPad Prism ver. 5.1 software. * *p* < 0.05, ** *p* < 0.01. # *p* < 0.05 between PPCs and POPCs.

**Figure 4 nutrients-14-03718-f004:**
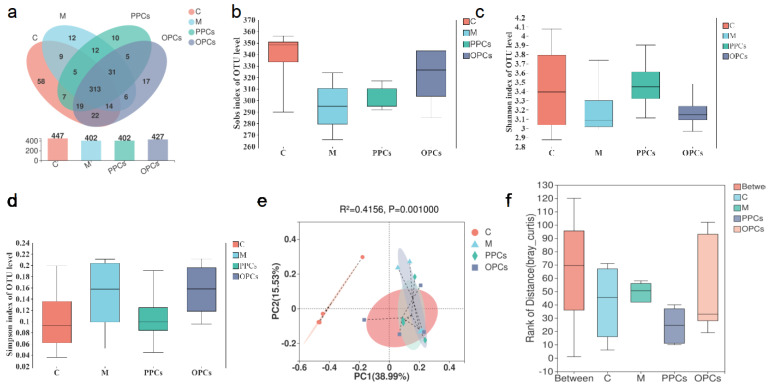
Gut microbiota composition of P-PCs and P-OPCs treatment mice**.** C57BL/6 mice were fed a HFD with the administration of P-PCs and P-OPCs. (**a**) OTU numbers of gut microbiota in different treatment groups; (**b**) Sobs indexes of OTU level; (**c**) Shannon indexes of OTU level; (**d**) Simpson indexes of OTU level; (**e**,**f**) PCoA analysis of the gut microbiota composition at OTU level.

**Figure 5 nutrients-14-03718-f005:**
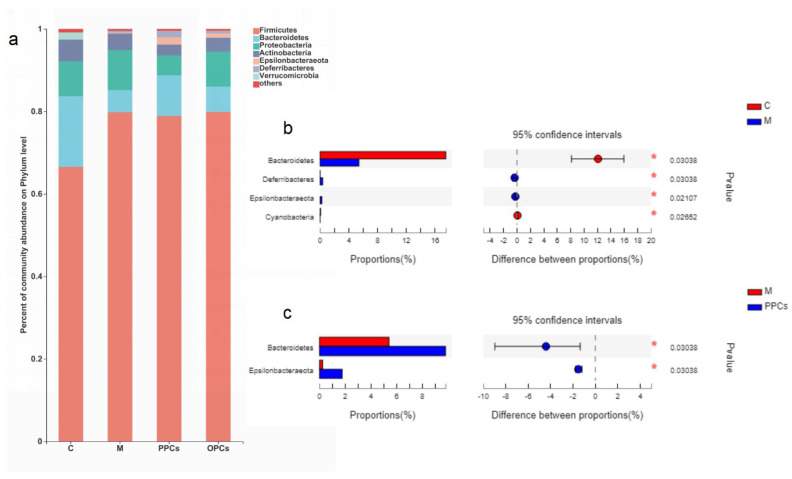
Gut microbiota composition at phylum level**.** (**a**) Relative abundance of gut microbiota at phylum level; (**b**) The differences of gut microbiota between C and M, (**c**) The differences of gut microbiota between M and PPCs. Data were analyzed by Wilcoxon rank-sum test, * *p* < 0.05.

**Figure 6 nutrients-14-03718-f006:**
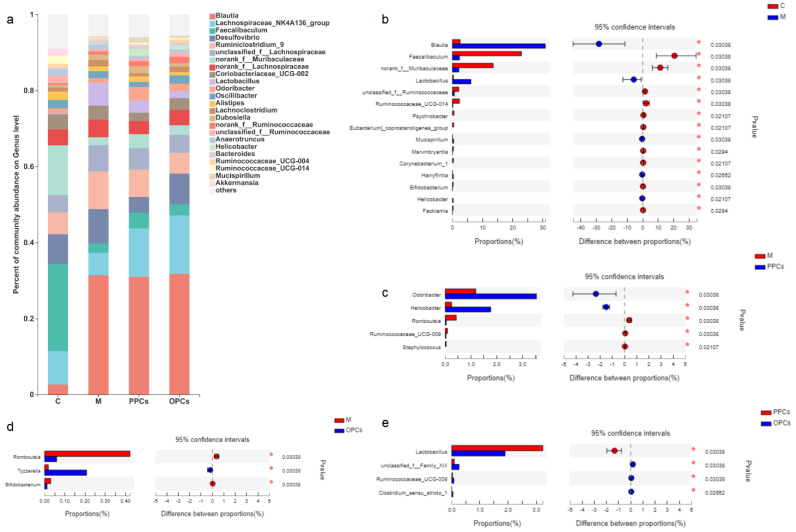
Gut microbiota composition at genus level**.** (**a**) Relative abundance of gut microbiota at genus level; (**b**) the differences of gut microbiota between C and M groups; (**c**) the differences of gut microbiota between M and PPCs groups; (**d**) the differences of gut microbiota between different M and POPCs groups; (**e**) the differences of gut microbiota between different PPCs and POPCs groups. Data were analyzed by Wilcoxon rank-sum test, * *p* < 0.05.

**Figure 7 nutrients-14-03718-f007:**
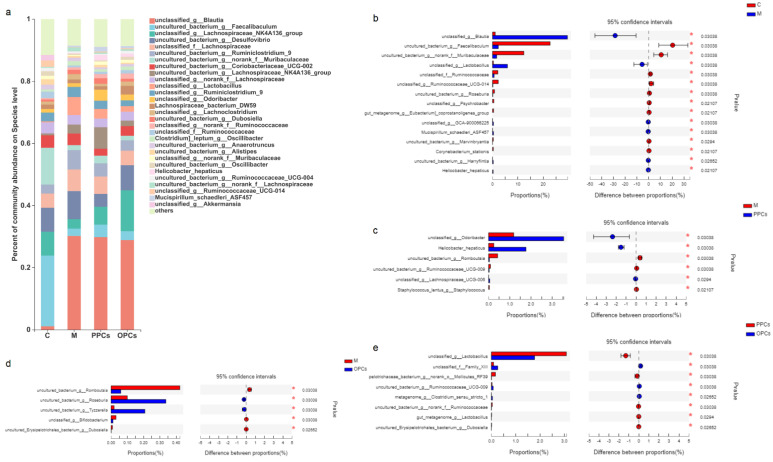
Gut microbiota composition at species level**.** (**a**) Relative abundance of gut microbiota at species level; (**b**) the differences of gut microbiota between C and M groups; (**c**) the differences of gut microbiota between M and PPCs groups; (**d**) the differences of gut microbiota between M and POPCs groups; (**e**) the differences of gut microbiota between PPCs and OPCs groups. Data were analyzed by Wilcoxon rank-sum test, * *p* < 0.05.

**Figure 8 nutrients-14-03718-f008:**
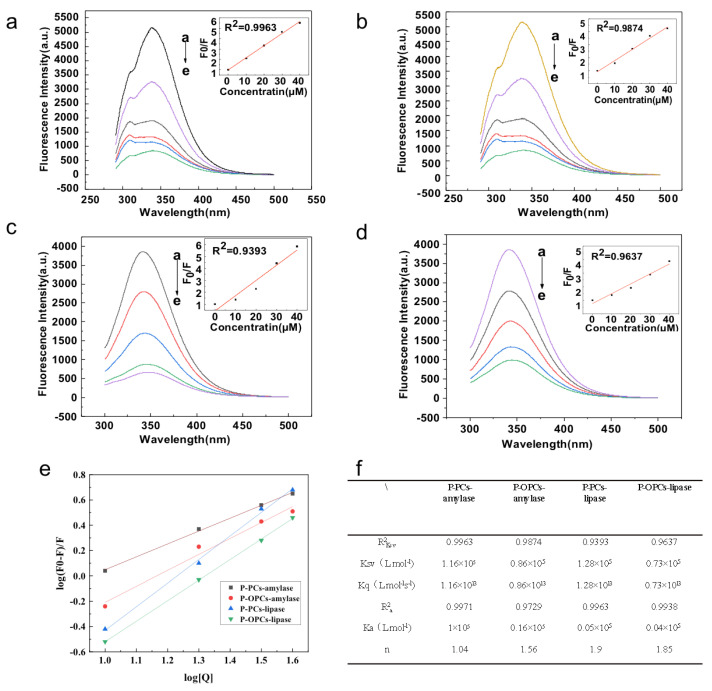
Fluorescence quenching spectra of Key Digestive enzymes at 37°C with P-PCs and P-OPCs. (**a**,**b**) Fluorescence quenching spectra of α-amylase at 37 °C with P-PCs and P-OPCs; (**c**,**d**) Fluorescence quenching spectra of lipase at 37 °C with P-PCs (**c**) and P-OPCs (**d**); (**e**,**f**) quenching constant Kq-values of binding between α-amylase and lipase at 37 °C with P-PCs and P-OPCs.

## Data Availability

All data generated or analyzed during this study are included in this published article. The data that support the findings of this study are available from the corresponding authors upon reasonable request.

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
