# Peer review of "Persimmon Proanthocyanidins with Different Degrees of Polymerization Possess Distinct Activities in Models of High Fat Diet Induced Obesity"

_nutrients, 2022, doi:10.3390/nu14183718_

Round 1

Reviewer 1 Report

The manuscript is well written with minor spelling and grammatical errors, which must be corrected.

However, there are too many non-standard abbreviations, which should be avoided. The reader will loose focus when degree of polymerization is abbreviated DP. Also HFD seem strange for high fat diet.  Nowadays there are no limitations to number of words in an article, so all words which have not any standard abbreviation should be written in full. And always write full word first -then abbreviation in parentheses.

The work is a continuation of previous projects form the authors, and the results data and the discussion of the results seem sound.

All discussed data from published articles should be included in the Introduction part, not only when project results are discussed.

Line 153: Figure 1: Text must be placed under the figure

Line 232: Figure 6: Text must be placed under the figure

Suppelementary table 1: Table text must be more precise, please describe the full content of the table.

Author Response

Thank you very much for reviewing our manuscript and also give us excellent suggestion. As you suggestion, we have revised the standard abbreviation in the manuscript, also the detail information for the legends of fig1,fig2 and table1 have been revised in the manuscript.

Reviewer 2 Report

In the manuscript “Persimmon proanthocyanidins with different degrees of polymerization possess distinct activities in models of high fat diet induced obesity” submitted to the Nutrients, the Authors investigated the potential effects of P-PCs and P-OPCs on high-fat diet induced obesity. The study was properly planned and conducted.

The only suggestion, which is not undermining the value of the study in any way, is that you missed Limitation of the work. Please also correct the conclusions. Conclusion must be relevant to the purpose of the work. The results should not be repeated in conclusions. 

Author Response

Thank you very much for reviewing our manuscript and also give us excellent suggestion. As you suggestion, we have carefully checked the manuscript and also revised the conclusion. please kindly find it in the revised manuscript.